# Associations between Ultrasonographically Diagnosed Lung Lesions, Clinical Parameters and Treatment Frequency in Veal Calves in an Austrian Fattening Farm

**DOI:** 10.3390/ani14162311

**Published:** 2024-08-08

**Authors:** Julia Hoffelner, Walter Peinhopf-Petz, Thomas Wittek

**Affiliations:** 1University Clinics for Ruminants, University of Veterinary Medicine Vienna, 1210 Vienna, Austria; thomas.wittek@vetmeduni.ac.at; 2PFI Dr. VET—The Veterinary OG, 8403 Lang, Austria; walter.peinhopf@dr-vet.at

**Keywords:** veal calf, bovine respiratory diseases, early detection, thoracic ultrasonography, physical examination, intranasal vaccine

## Abstract

**Simple Summary:**

Early diagnosis of lung lesions in calves is crucial for prompt treatment intervention and management of respiratory diseases. Both a physical and a transthoracic ultrasonography scoring at time of entry offers the opportunity to re-evaluate action plans in case of illness and further facilitate implementation of prophylactic management measures e.g., intranasal vaccination. The present study highlights the advantages of an early diagnosis of respiratory diseases on various production parameters of veal calf fattening and offers the opportunity to evaluate the effect of an intranasal vaccine for bovine respiratory diseases (Bovalto^®^ Respi Intranasal).

**Abstract:**

This study evaluated the significance and predictive value of ultrasonographic and physical examination on arrival at an Austrian fattening farm. Treatment frequency and average daily weight gain (ADG) were related to physical and ultrasonographic examination results. Additionally, the effect of an intranasal vaccination in half of the examined calves was studied. The clinical and ultrasonographic health status 600 calves was recorded at the beginning and end of fattening. Half of the calves received an intranasal vaccination (Bovalto^®^ Respi Intranasal). Overall, 44.5% showed an abnormal respiratory scoring (RS) and 56.0% showed signs of respiratory diseases in transthoracic ultrasonography (TUS) at arrival on the farm. For both RS and TUS, a categorization between ILL and HEALTHY was conducted. Results showed lower ADG in ILL calves (RS median: 0.93 kg/d; TUS median: 0.96 kg/d) compared to HEALTHY calves (RS median: 1.01 kg/d; TUS median: 1.01 kg/d). The median ADG was lower in not treated and ILL calves (RS median 0.90 kg/d; TUS: 0.93 kg/d) compared to treated and ILL calves (RS median 1.01 kg/d; TUS: 1.02 kg/d). Vaccination did not affect growth performance or occurrence of ILL, though treatment frequency was lower in VAC calves (17.0% in NVAC; 11.3% in VAC). The implementation of examination protocols for respiratory diseases may have a positive impact on production parameters (e.g., treatment frequency and ADG).

## 1. Introduction

The veal calf production system in Austria is mainly based on fattening of dual-purpose breed (Simmental) or crossbred calves but also of so-called “surplus calves” of dairy breeds (Holstein, Brown Swiss). The dairy calves are mainly males but also include females that are not needed for herd replacement [1,2]. Due to their lower economical value for the dairy farmers, these calves might not be reared equally (e.g., worse colostrum, feeding, and housing management) than breeding calves for herd replacement at the farms of origin [3,4]. This, in combination with poor immune status (immunological gap) and stress (transport to the market and fattening farm, new housing, and feeding conditions) results in a high risk for morbidity at the fattening farm [1]. The aim to minimize infectious diseases and an increased emphasis on animal welfare influenced the management of veal calf production over time. A healthy growth of veal calves starts with appropriate colostrum management and vaccination at the farm where the calves are born [5] and continues with further raising of fattening calves in a controlled hygienic stress-reduced environment with suitable diet and feeding schedules [6]. These management procedures aim to minimize the risk of infectious diseases like bovine respiratory disease (BRD).

BRD is one of the most frequent diseases in veal calves and often requires treatment including antimicrobial drugs. It is still a frequently used procedure to treat all calves with antimicrobial substances at the arrival in the fattening farm. This contributes to the development of antimicrobial resistance (AMR) [7]. In several studies, an increase of AMR was reported for major BRD pathogens (*Mannheimia haemolytica*, *Pasteurella multocida*, and *Histophilus somni*) [8,9,10,11,12]. Interactions between different pathogens, hosts, drugs, and environments may result in treatment failures. An evaluation of treatment protocols and diagnostic results is preferable, as a simple change of antimicrobial substance is often not successful [13]. 

An appropriate vaccination program post-arrival at the fattening farm can be an alternative method to reduce occurrence of BRD without the risk for development of antimicrobial resistance [14]. However, it would be preferable to vaccinate calves prior to shipment to the fattening farm so that the calves are already vaccinated when they arrive at the farm. Well-designed vaccination programs can reduce the prevalence of disease and can protect unvaccinated or vulnerable individuals within the herd [15,16]. In veal calf production, intranasal vaccines are often implemented in disease management protocols because of their practicability and their possible application in newborn calves within the first week of life. A single intranasal administration with a rapid onset of effect improves user convenience and facilitate administration [15,17]. 

Advanced diagnostic procedures, for instance transthoracic ultrasonography (TU), provide detailed information on ongoing disease processes [18]. The lung ultrasonography allows an objective, measurable, and quantifiable diagnosis of lung lesions and can be integrated in calf health monitoring systems without any major effort [18,19]. Further, TU comes closer to the research objective of finding a gold standard method of in vivo diagnosis of BRD. The identification of lung lesions using TU and its association with pneumonia was proved by accuracy tests in several studies. High values of sensitivity and specificity were generated in Rabeling et al. (1998) (Se: 0.85%; Sp: 98%), in Buczinski et al. (2015) (Se: 79.4%; Sp: 93.9%), and Ollivett et al. (2015) (Se: 94%; Sp: 100%) [15,20,21]. In combination with clinical parameters (nasal discharge, eye discharge, abnormal breathing, body temperature, head tilt, or ear drop). a differentiation between four subtypes of respiratory disease is possible: “Healthy”, meaning no ultrasonographic or clinical signs of BRD, “infection of the upper respiratory tract” excluding ultrasonographic signs of lung diseases, “subclinical pneumonia” excluding clinical signs of respiratory diseases, and, in cases of acute pneumonia, ultrasonographic and clinical signs of respiratory disease can be found [19]. Early identification and a clear allocation of respiratory diseases can maintain individual therapy decisions (no antibiotic treatment in case of healthy or upper respiratory tract infection, early and consequent antimicrobial therapy in case of pneumonic lesions). The aim is to increase farm productibility, animal well-being, and take care of “prudent use of antibiotics” [19]. 

Additionally, the effects of management changes (e.g., changes of treatment protocols, in barn climate conditions, or implementation of vaccination programs) on calf respiratory health can be evaluated using thoracic ultrasonography over a period of time [19,22]. Monitoring of treatment records in the context of prior assessment of lung consolidation is especially important when evaluating therapy effectiveness. Early antimicrobial therapy can immediately limit the progression of lung consolidation [23]. A study of Buczinski et al. (2014) pointed out that only 41.1% of calves showing ultrasonographical signs of lung consolidation were previously treated, which highlights the importance of early antimicrobial treatment in cases of lung diseases to avoid negative consequences on various production parameters (e.g., a lower average daily weight gain) [24,25].

In several studies, a lower average daily weight gain (ADG) was recorded if a lung consolidation ≥ 1 cm^2^ was found [26,27]. Nevertheless, the association between ultrasonographic lung lesions and specific production parameters (ADG, body weight at arrival on the fattening farm, slaughter weight) is complex and may depend on individual parameters (sex, breed, age, immune status) and farm individual management procedures (vaccination, treatment, feeding regime, barn ventilation). 

The first objective of this study was to evaluate the association between ultrasonographically and clinically assessed health status of veal calves, production parameters (ADG, arrival weight, slaughter weight, sex, breed, age), and treatment frequency. The second objective was to assess the effect of an intranasal vaccination (Bovalto^®^ Respi Intranasal) on ultrasonographic and clinical health as well as growth and production parameters of veal calves. Two hypotheses were formulated:

First: ADG and treatment frequency are associated with results of the first physical and ultrasonographic examination at the start of fattening. 

Second: The intranasal vaccination Bovalto^®^ Respi Intranasal positively influences the daily weight gain, treatment frequency, and TUS and RS. 

## 2. Materials and Methods

### 2.1. Animals and Housing

This study was performed on a fattening farm in Austria between March 2021 to March 2022. The farm regularly buys 30 to 80 calves (amount depending on the market situation) from three different calf auctions: two in the Austrian federal state Styria and one in Austrian federal state Salzburg. No detailed information about farm of origin (size, types of farming, feeding, or distance of transport to the calf markets) is available for the fattening farm. However, the control of transport papers and a mandatory veterinary examination ensures a sufficient health status and provides additional information about each calf for the purchaser. Following a quarantine phase, during which calves were housed in individual boxes for 5–7 days, calves were housed in separate small groups of six calves. 

All disease diagnoses and treatments were routinely recorded in the farm electronic documentation system. In case of clinical illness, a standardized treatment protocol was used, which had been implemented by farm staff and farm veterinarian. Clinically ill calves were recorded and treated by farm staff independently from the study in accordance with the farm veterinarian. Due to market constraints, the withdrawal period of antimicrobial products had to be as short as possible (AMA-quality label: doubles withdrawal time).

In case of BRD during fattening a combined therapy with antimicrobial substances (e.g., cefquinome, amoxicillin, or oxytetracycline) and meloxicam, ketoprofen, or flunixin-meglumine *transdermal* was used. Individual therapy decisions were based on regularly collected nasal swabs and simultaneous recommendations to improve air conditions (e.g., tube ventilation). In suspected cases of Mycoplasma, tylosin was administered. All calves of one unit received antimicrobial therapy (amoxicillin, doxycycline) at arrival *per os* for a duration of seven days. Additionally, group therapy was preferred to individual therapy if more than 50% of one group showed clinical signs of respiratory disease (eye or nasal discharge, increasing breathing frequency, ear droop or head tilt, or an increased body temperature ≥ 39.3 °C). All medical therapy procedures were supported by accompanying measures: improvement of hygienic standards, cleaning calf boxes more frequently, and closer health checks at purchase. Veterinary advice was additionally consulted in case of severe cases and a loss of appetite. 

Alternatingly, calves were assigned to two treatment groups: 1. The intranasal vaccination group (VAC) and 2. The non-vaccinated Placebo-Group (NVAC). In one week, all calves were allocated to the VAC group, whereas in the following week, all calves were in the NVAC group. The dose of 2 mL of the vaccine (Bovalto^®^ Respi Intranasal, Boehringer Ingelheim) was administered intranasally (1 mL per nostril) with a single use plastic applicator to the calves of the VAC groups. The calves in the NVAC groups received isotonic saline solution; the application method and volume did not differ from the application of the vaccine. 

In this study, 600 calves of different breeds and sexes were enrolled. Calves had an average body weight of 91 kg (minimum: 60 kg; maximum: 130 kg) at arrival. This study was conducted as a field trial that also compared the effect of intranasal vaccination after arrival at the farm. The initial expected prevalence for respiratory disease was 30%. A sample size of 300 calves per treatment group was calculated using power analysis (95% confidence level, a standard deviation of 0.5, margin of error ± 5%). 

The study was discussed with the institutional ethics and welfare committee and approved by the Austrian Federal Ministry for Education, Science and Research (GZ 2020-0.773.262) in accordance with Good Scientific Practice guidelines and national legislation.

### 2.2. Data Collection Ante Mortem

Data of respiratory health status, using respiratory scoring (RS) and transthoracic ultrasonography (TUS), were collected twice during the fattening period. The first examination took place two days after arrival at the farm, while the second examination took place on the day of slaughter. The TUS examination protocol and the physical examination (following the RS system of pre-weaned dairy calves of UC Davis) was applied by the primary author (JH) [18]. 

Respiratory function scoring was conducted by a modified system of McGuirk and Peek (2014) [28]. Six clinical signs, associated with respiratory system and function, cough (CO), nasal discharge (ND), eye discharge (ED), increased breathing frequency (BR), head tilt or ear droop (HAT), Temperature (T), and Auscultation (AU), were assessed during physical examination. This scoring system is based on allocation of calves showing “no symptoms” (=0 points) and calves showing “symptoms” irrespective of the severity. They were scored with 2 (CO, ED, BR, T), 3 (AU), 4 (ND), or 5 (HAT) points. The AU was compared between “no symptom” (=normal breathing symptom) = 0 points and “symptoms” (crackles, wheezes, stridor, or an absence of breathing symptoms) = 3 points [29,30].

The examination procedure of TUS was conducted using a modified examination procedure of Ollivett et al. (2016) and has been used by the authors before [18,19]. A linear probe (7.5 MHz) was moved from dorsal to ventral between every intercostal space beginning on right side between the 10th intercostal space (caudal landscape: diaphragm). Every intercostal space was systemically scanned until the second ICS (cranial landscape: heart). 

For ultrasonographic examination, a linear probe of a rectal ultrasound unit (Tringa Linear Vet, Esaote, Genova, Italy) was used. Calves were unshaved and 70% isopropyl alcohol was used as the transducing agent [18,19]. According to the examination protocols, calves with ≥4 points out of 24 points were considered as “ILL”. Based on four findings (presence of COMTs and Consolidation, Atelectasis, Alveolograms, and Fluid), a scoring system was generated for each of the four lung areas (L1, L2, R1, R3). COMT were scored from 0–3 depending on the size of consolidated lung tissue: 0 = No COMTs, normally aired/ventilated, no consolidation; 1 = More than five COMTs, diffusely allocated, no consolidation; 2 = More than ten COMTs, small lobular lesions; 3 = Lobar lesion of the whole lobe. Alveolograms, atelectasis, and fluid were scored with 0 or 1 points, depending on the presence (1) or absence (0) of the mentioned diagnosis. 

### 2.3. Data Collection Post Mortem

After a fattening period of approximately 77 days (minimum: 30 days; maximum: 100 days), calves were transported to the nearby abattoir (Weiz, Austria). Depending on demand, five to thirty calves were slaughtered two days per week. The observer (JH) was able to examine the lungs of each calf for approximately 60 s, integrated in the *postmortem* inspection process. Results were recorded on paper and later transferred into an Excel spreadsheet. The *postmortem* scoring system of Leruste et al. (2012), which ranges from 0–3 points, was used (0 = healthy lung with a normal pale orange color; 1 = one spot of grey-red discoloration; 2 = one larger or several small spots of grey-red discoloration; 3 = grey-red discoloration area of a full lobe and/or presence of abscesses), and it was applied for each of the five lung areas (Figure 1) [31]. 

### 2.4. Statistical Analyses

Raw data were transferred to Microsoft Excel (version 2310, build 16.0.16924.20054, Microsoft Corporation, Redmond, WA, USA) and all parameters for statistical analyses (+tables and figures) were calculated in Excel. 

For statistical analyses, calves were classified in two categories: ILL and HEALTHY. For both examination scores (RS and TUS), a cut-off point was calculated. HEALTHY relates to a low score, where no severe lung lesions or only mild clinical signs could be detected. Calves that are categorized as ILL show severe lung lesions or increased clinical signs. For RS, the cut-off point was calculated at 7 points (calves ≥ 7 points are ILL and calves < 7 points are HEALTHY). For TUS, the cut-off point was calculated at 4 points (calves ≥ 4 points are ILL and calves < 4 points are HEALTHY) [18].

To evaluate significant differences between the proportion of sex, breed, and frequency of treatments, the Chi-square independence test was used. Significant results were generated at a significance of *p* < 0.05 (critical value 3.84 and significance level α 0.05). 

All quantitative variables were tested for normal distribution (Shapiro–Wilk test). A normal distribution of data was found for ADG and slaughter weight. Results for fattening duration were not proved to be normally distributed.

For detection of significant differences between the median ADG of clinically/ultrasonographically ILL and HEALTHY calves of the first examination at the start of fattening, the Mann–Whitney U-Test was used. *p* < 0.05 was considered indicative of statistical differences. Differences of median ADG in treated and not treated ILL calves were calculated equally (Mann–Whitney U-Test, *p* < 0.05). 

## 3. Results 

In the examination methods, “physical examination” and “transthoracic ultrasonography” HEALTHY and ILL calves were categorized according to cut-off points. In the “*postmortem* inspection”, absence or presence of lung lesions led to the categories HEALTHY and ILL. The two groups of each examination method were compared in terms of treatment against BRD and ADG. 

### 3.1. Physical Examination

Of a total of *n* = 600 calves, physically ILL calves were detected most frequently during the examination at the beginning of fattening (RS1), where *n* = 267 (44.5%), followed by examination at the end of fattening (RS2), where *n* = 224 (37.3%). Significant differences between ILL and HEALTHY calves could be found in all categories (ADG, breed, and treatment frequency) except sex (Table 1). In general, calves from all examination groups were more frequently female and crossbreeds. Additionally, for both examinations (RS1 and RS2), ILL calves were treated more frequent and had a lower ADG than HEALTHY calves.

### 3.2. Transthoracic Ultrasonography (TUS)

The proportion of ultrasonographically ILL calves at first examination at the beginning of fattening (TUS1) was 56.0% (336 out of 600). Compared to results in TUS2 at the end of fattening, the proportion was lower (48.3%, *n* = 290). Respectively, HEALTHY calves showed similar results (44%, *n* = 264 in TUS1; 51.7%, *n* = 310 TUS2). Additionally, significant differences can be found between ultrasonographically ILL and HEALTHY calves in all parameters (treatment frequency, ADG, sex, breed) (Table 2). Significant differences between treatments (TUS1 and TUS2) could only be calculated between the total amount of treated and not treated calves (for both ILL and HEALTHY). Female and crossbreed calves were more frequently diagnosed ultrasonographically as ILL. 

### 3.3. Post Mortem Inspection of the Lungs

A total of 210 (35.0%) calves showed lung lesions (moderate to severe) at slaughter. The distribution of treatment frequency, sex, breed, and ADG between ILL and HEALTHY calves is presented in Table 3. No difference could be detected between the ADG of ILL and HEALTHY calves. Female and crossbreed calves were more likely ILL than HEALTHY. ILL calves were treated more frequently (two or more times) than HEALTHY calves. Due to high rates of HEALTHY calves (65.0%) compared to ILL calves (35.0%), the number of calves that were treated once was higher in HEALTHY calves. 

### 3.4. Association of the Examination Results to ADG

In Figure 2, a significant difference between ADG (kg/d) for clinically ILL (44.5%; *n* = 267) and HEALTHY (55.5%; *n* = 333) calves is demonstrated. A significant difference (*p* < 0.05) was calculated at a median of 0.93 kg/d for physically ILL calves compared to a median of 1.01 kg/d for physically HEALTHY calves. The maximum ADG is 1.62 kg/d (for ILL) and 1.70 kg/d (for HEALTHY) compared to a minimum ADG of 0.16 kg/d (for ILL) and 0.18 kg/d (for HEALTHY). 

Figure 3 shows the difference between ADG (kg/d) for ultrasonographically ILL (56.0%; *n* = 336) and HEALTHY (44.0%; *n* = 264) calves. A significant difference was calculated at a median 0.96 kg/d for ultrasonographically ILL calves compared to a median of 1.01 kg/d for ultrasonographically HEALTHY calves. The maximum was 1.62 (for ILL) and 1.70 (for HEALTHY).

### 3.5. Influence of Treatment Frequency in Clinically and Ultrasonographically ILL Calves

#### 3.5.1. Clinically ILL Calves

The ADG of non-treated and treated calves was calculated for clinically ILL calves (Figure 4). A significant difference was determined between the ADG (median 1.01 kg/d) of treated clinically ILL calves (26.7%; *n* = 73) and for not treated clinically ILL (72.7%; *n* = 194) calves (ADG median 0.90 kg/d). Clinically ILL calves that were not treated had a minimum ADG of 0.16 kg/d compared to a minimum ADG of 0.23 kg/d for clinically ILL and treated calves. The maximum ADG was 1.57 kg/d for treated and 1.62 kg/d for not treated calves. 

#### 3.5.2. Ultrasonographically ILL Calves

Figure 5 shows the ADG of non-treated and ultrasonographically ILL calves at TUS1 at the beginning of the fattening period. The median of treated ultrasonographically ILL calves (23.8%; *n* = 80) was 1.02 kg/d and for not treated ultrasonographically ILL (76.2%; *n* = 256) calves 0.93 kg/d, which was a significant difference (*p* < 0.05). Treated calves that were detected as ultrasonographically ILL had a higher ADG than not treated calves and have a minimum ADG of 0.23 kg/d and a maximum ADG of 1.57 kg/d. The minimum ADG for not treated ILL calves was 0.16 kg/d and the maximum was 0.62 kg/d.

### 3.6. Vaccination

A total of 600 calves were enrolled and randomized into two groups (VAC, *n* = 300); NVAC, *n* = 300) and studied for differences between the groups considering sex, breed, and examination results at the beginning of the fattening period (Table 4). Between VAC and NVAC, no significant differences could be found except for breed. 

After controlling the effects of calf individual characteristics, frequency of treatments, ADG, fattening duration, and results of second examination at the end of fattening were compared between VAC and NVAC calves (Table 5). In total, 28 calves were treated at least once during the fattening and 6 were treated twice. Compared to NVAC, VAC calves were less often treated and the disease frequency in cases of illness was higher in NVAC than in VAC calves. However, significant differences could only be calculated for the total amount of treatments and calves treated a maximum of five times. In VAC calves, zero calves were treated five times due to respiratory disease compared to five calves that were included in the NVAC calves’ group. When comparing ILL and HEALTHY calves in the VAC and NVAC groups, no significant differences could be found for RS2 in VAC (ILL calves: *n* = 108; HEALTHY calves *n* = 192) and NVAC (ILL *n* = 116; HEALTHY *n* = 184).

## 4. Discussion

In veal calf production, respiratory diseases have a major impact on calf health performance and directly influence the economic profitability of a farm. Due to the complexity of disease-causing factors and the high variability of disease related clinical signs, the implementation of disease control programs and risk assessment tools for early detection of BRD is gaining importance [33]. To the authors knowledge, this is the first study analyzing data of production parameters (e.g., sex, breed, ADG, treatments) and their association with physical and ultrasonographic respiratory health status on an Austrian fattening farm. The implementation of a routine on-farm usage of physical RS and TUS at the beginning of fattening was evaluated using data of ADG and treatments. Additionally, the preventive value of an intranasal vaccination was tested and evaluated in context of physical and ultrasonographic health status and various productive parameters under field conditions.

### 4.1. Association between Clinical and Ultrasonographic Respiratory Health Status and ADG, Treatment Frequency, Sex and Breed at the Beginning of Fattening

Aly et al. (2020) [33] pointed out that the components for a risk assessment tool are based on exact evaluation of risk factors for BRD (calf individual parameters, housing, and management) and continuous records about the progression of the disease. Both a detailed risk factor questionnaire and the California BRD scoring system were found to be reliable tools to implement herd-specific prevention and control strategies for BRD [33]. The study implemented transthoracic ultrasonography as an additional examination method for detection of lung lesions. In several studies, a moderate to high sensitivity and high specificity are frequently declared for this examination method [15,18,19,34]. However, the TU examination did not prevail as the gold standard reference method for in vivo diagnosis of BRD. The difficulty of differentiating between “treatment-worthy” active cases of pneumonia and not “treatment-worthy” lesions following a previous lung disease exclude TU as the sole method of in vivo BRD diagnosis so far [24]. 

The general economic impact of BRD in the cattle industry is measured by its influence on ADG, treatment costs, and mortality. Calves diagnosed with BRD (RS and TUS) had lower ADG compared to healthy animals. Over an average fattening period of 76 days, a difference of 0.08 kg/d could be detected between ILL and HEALTHY calves in RS and 0.05 kg/d in TUS. In our opinion, results of treatment frequency should be included alongside results of development of ADG as a parameter for respiratory diseases during the fattening period because treatment of previously detected ill calves (scored in RS and TU) has proved to have a significant positive effect on ADG (1.02 kg/d treated and ill calves versus 0.93 kg/d not treated and ill calves). These results coincide with several previous studies [34,35,36,37] and emphasize the importance of early detection of respiratory diseases. In addition to positive effects on animal health and welfare (early treatment of diseased calves) and a positive impact on farm economics (positive development of ADG), the benefits of an early therapeutical intervention should also be taken into consideration. An early determination of respiratory health status followed by appropriate antimicrobial treatment may lead to a compensatory gain of weight by the end of fattening [38]. 

However, using “treatment frequency” for singular characterization of respiratory health status of a farm is questionable because higher treatment rates were not found in calves that died during the fattening period [39,40] nor in calves showing lung lesions (independently of degree and severity) at slaughter [41,42]. Only the differentiation between severe and minor lung lesions has the advantage of detecting diseased calves because of previous treatment. Severely diseased calves were more likely to be treated than calves without lung lesions [43]. These findings are consistent with our study showing higher treatment frequency in calves diagnosed as ILL (23.8% in TUS, 27.4% in RS) on the arrival at the farm compared to HEALTHY (1.9% in TUS, 3.6% in RS). Further, we conclude that a reduction of antimicrobial treatment during fattening (targeted antimicrobial treatment at the day of entry) is possible if treatment decisions are based on prior assessment of respiratory health status (RS and TUS). 

The proportion of female and male calves showing signs of respiratory diseases was higher in male calves for both examinations (RS and TUS). Statistical significance was generated in TUS1 where female calves were more likely to be classified as HEALTHY than ILL (63.4% female and ILL compared to 74.2% female and HEALTHY) and male calves were more likely to be classified as ILL than HEALTHY (36.6% male and ILL and 25.8% male and HEALTHY). For preweaned calves, a failure of passive immune transfer, as discussed in Aly et al. (2020), may explain the high disease incidence for BRD [25]. For our study purpose, a detailed analysis of calf management procedures to evaluate the association between feeding procedures of colostrum and failure of passive immune transfer on the company of origin was not implementable because of the high labor intensity and the high variability of housing and management conditions in small-scale farm structures. 

Further, analysis of breed revealed higher percentages of purebred calves that were classified as ILL for BRD (RS and TUS) compared to crossbred calves (they were more often classified as HEALTHY than ILL). We conclude that a higher economic value of crossbred calves (for producers) may result in higher attention to calf health issues in preweaned crossbred compared to purebred calves. Nevertheless, previous studies showed a low heritability of resistance to BRD. The influence of disease incidence on heritability is discussed: the higher the disease incidence, the more calves could express phenotypes for disease resistance [44,45].

### 4.2. Influence of Intranasal Vaccination on Respiratory Health Status, ADG and Treatment Frequency

Our study results are partially in contrast to several previous studies, where an improved respiratory health status could be found using different vaccination protocols (parenteral and intranasal) against BRD [46,47]. No significant differences between clinical or ultrasonographical health status and vaccination groups (placebo or intranasal vaccination) could be detected at the end of fattening. While Sandeline et al. (2020) and Ollivett et al. (2018) proved the efficacy of an intranasal vaccination by a positive effect on growth performance and fewer lung lesions at slaughter [22,47], a decreasing treatment frequency was the only factor to significantly differ between VAC and NVAC calves in this study (17.0% of NVAC and 11.3% of VAC calves were treated because of respiratory diseases). For the average fattening duration of 76 days, the ADG in both VAC and NVAC calves do not significantly differ (0.98 kg/d in VAC; 0.97 kg/d for NVAC). The most likely reason is the time of vaccination after the calves has been transported to the farm, which is late in comparison to other studies. Vaccination before transport (ideally between 10–14 days of age) might reduce the high rate of BRD positive cases at the beginning of fattening. This may result in more beneficial effects of vaccination over the short study period. 

Further, we assumed an overlapping herd immunity because of cohousing vaccinated and not vaccinated calves, also described in Iman and Hudson (2009) [48]. To ensure the good validity of our study, we followed the guidelines for vaccine studies under field conditions, which recommends no separation between vaccination groups to ensure similar environmental (exposure to pathogens) conditions [49]. Additionally, a homogenous distribution of calf characteristics (small variation of breed, sex, fattening duration, and physical and ultrasonographic health status within the groups) could be detected, however these do not reflect significant differences in ADG between VAC and NVAC calves. The usage of metaphylactic antimicrobial treatment must be considered when interpreting the study results. The limitation of colonialization of bacteria can positively influence the defense mechanism of the respiratory tract against pathogens [50,51]. Thus, a combination with good housing and ventilation conditions might ameliorate growth performance, even for NVAC calves. Therefore, the comparison between VAC and NVAC calves is influenced by other factors, such as the antimicrobial treatment. 

### 4.3. Limitations

Although differences between groups of calves (ILL and HEALTHY) were noted, the fact that only one fattening farm was included is a limitation when seeking to draw general conclusions about respiratory diseases and the development of production parameters in general. An inclusion of more herds and a higher number of calves might lead to a higher variance of respiratory health status results. The authors also acknowledge that the study design was not perfectly blinded. However, the feasibility and personal expenses necessary for additional examinations were considered impractical. The authors also acknowledge that additional examination methods (serological tests, bacterial culture) to test the efficacy of the intranasal vaccine were not available. A lack of serological tests for verification of the presence of vaccine-induced antibodies against pathogens weakens statements concerning the protective value of the vaccine [52]. However, these are the typical limitations of a field study that includes a rather high number of animals.

## 5. Conclusions

Overall, our findings provide additional information about the contribution of host components to BRD in veal calf production. Appropriate diagnostic tools in combination of continuous monitoring of calf health, especially at the time of entry, seems to be an effective strategy to reduce negative consequences on important production parameters (e.g., ADG and treatment). 

Although preventive measurements to reduce the risk of an BRD outbreak on a farm are highly important, efficacy of e.g., intranasal vaccination, is variable. While disease causing treatments were reduced in vaccinated calves, no benefits on health status or ADG could be generated. This study needs to be confirmed with a larger number of herds and the inclusion of serological tests to improve the informative value.

## Figures and Tables

**Figure 1 animals-14-02311-f001:**
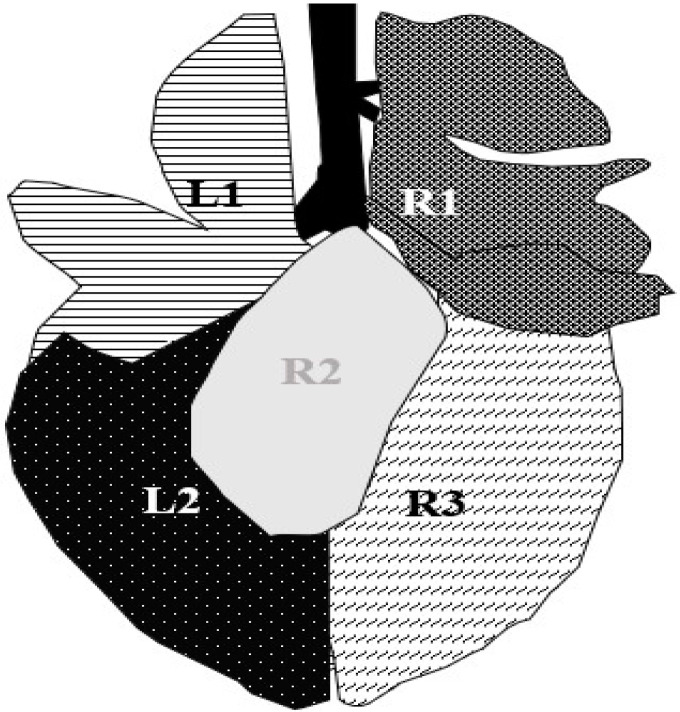
Segmentation of calves’ lungs in five examination areas. L1 = lobus cranialis sinister, L2 = lobus caudalis sinister, R1 = lobus cranialis dexter + lobus medialis dexter, R2 = lobus accessories (accessible in slaughter examination exclusively), R3 = lobus caudalis dexter [18,32].

**Figure 2 animals-14-02311-f002:**
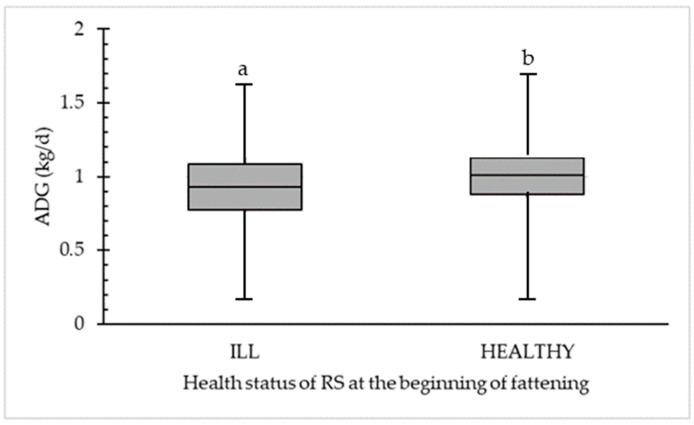
Significant difference in average daily weight gain (ADG in kg/d) between clinically ILL (*n* = 267) and HEALTHY (*n* = 333) calves were detected in the first respiratory scoring (RS) at the beginning of fattening. Different letters (a,b) within the category (ILL or HEALTHY) are statistically different (*p* < 0.05).

**Figure 3 animals-14-02311-f003:**
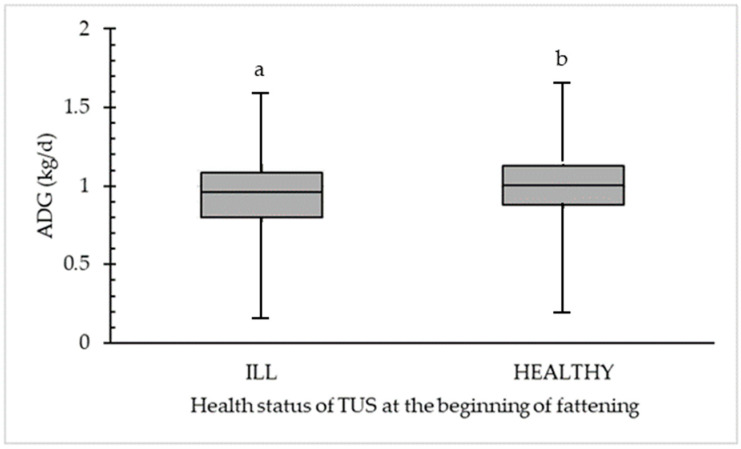
Significant difference in average daily weight gain (ADG in kg/d) between ultrasonographically ILL (*n* = 336) and HEALTHY (*n* = 264) calves detected in the first transthoracic ultrasonography scoring (TUS1) at the beginning of fattening. Different letters (a,b) within the category (ILL or HEALTHY) are statistically different (*p* < 0.05).

**Figure 4 animals-14-02311-f004:**
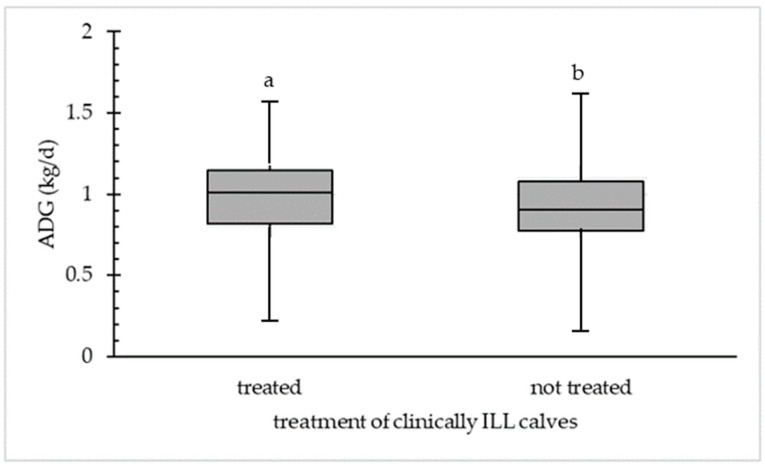
Difference in average daily weight gain (ADG in kg/d) in treated (*n* = 73) and not treated (*n* = 194) calves detected in clinically ILL calves at first physical examination at the beginning of fattening. Different letters (a,b) within the category (treated or not treated) are statistically different (*p* < 0.05).

**Figure 5 animals-14-02311-f005:**
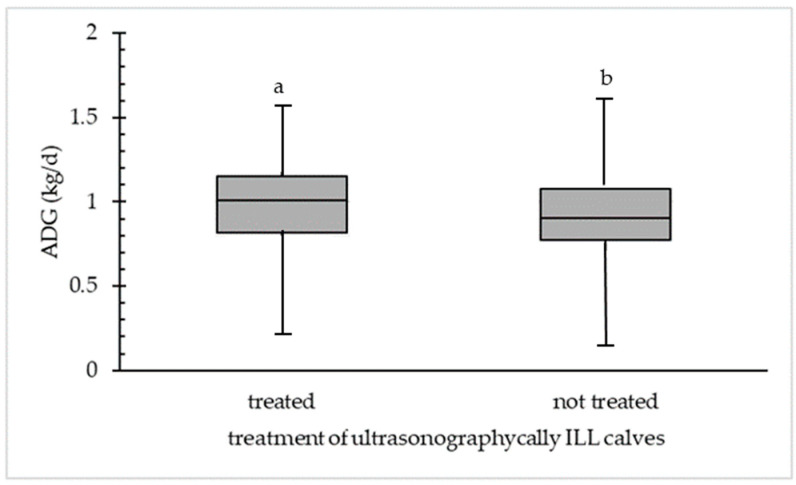
Difference in average daily weight gain (ADG in kg/d) between treated (*n* = 80) and not treated (*n* = 256) ultrasonographically ILL calves at first ultrasonographic examination at the beginning of fattening. Different letters (a,b) within the category (treated or not treated) are statistically different (*p* < 0.05).

**Table 1 animals-14-02311-t001:** Distribution of various parameters in physically ILL ^1^ and HEALTY ^2^ calves in numbers (%). To evaluate differences between the two examinations at the start and end of fattening, calves (*n* = 600) were divided into two groups: RS1 ^3^, RS2 ^4^.

		RS1 ^3^ (*n* = 600)		RS2 ^4^ (*n* = 600)	
Variable	Category	ILL ^1^ *n* = 267 (44.5) ^A,b^	HEALTHY ^2^ *n* = 333 (55.5) ^B,b^	ILL ^1^ *n* = 224 (37.3) ^A,a^	HEALTHY ^2^ *n* = 376 (62.6) ^B,a^
Frequency of treatments	0	194 (72.7) ^A^	321 (96.4) ^B,b^	170 (75.9) ^A^	345 (91.8) ^B,a^
	1	53 (19.9) ^A^	12 (3.6) ^B,b^	36 (16.1) ^A^	29 (7.7) ^B,a^
	2	14 (5.2) ^A^	0 (0.0) ^B^	13 (5.8) ^A^	1 (0.3) ^B^
	3	1 (0.4)	0 (0.0)	1 (0.4)	0 (0.0)
	4	5 (1.9) ^A^	0 (0.0) ^B^	4 (1.8) ^A^	1 (0.3) ^B^
	total	73 (27.4) ^A,b^	12 (3.6) ^B,b^	54 (24.1) ^A,a^	31 (8.2) ^B,a^
sex	male	94 (35.2)	97 (29.1)	84 (37.5)	107 (28.5)
	female	173 (64.8)	236 (70.9)	140 (62.5)	269 (71.5)
breed	purebred	121 ^A^ (45.3)	123 ^B^ (36.9)	94 (42.0)	145 (38.6)
	crossbred	146 ^A^ (54.7)	210 ^B^ (63.1)	125 (55.8)	231 (61.4)
ADG	Median (1st and 3rd Quartile)	0.93 (0.78; 1.08) ^A^	1.01 (0.88; 1.13) ^B^	0.94 (0.76–1.11)	1.00 (0.88–1.11)

^1^ Physically ILL calves with a total score ≥ 7 points. ^2^ Physically HEALTHY calves with a total score < 7 points. ^3^ Physically ILL/HEALTHY calves of the first examination at the start of fattening (RS1). ^4^ Physically ILL/HEALTHY calves of the second examination at the end of fattening (RS2). ^A,B^ Alphabetical upper-case indices indicating significant differences between ILL (A) and HEALTHY (B) calves at *p* < 0.05. ^a,b^ alphabetical lower-case indices indicating significant differences between ILL calves of RS 1 (a) and RS2 (b) and significant differences between HEALTHY calves of RS1 (a) and RS2 (b).

**Table 2 animals-14-02311-t002:** Distribution of various parameters in ultrasonographically ILL ^1^ and HEALTHY ^2^ calves in numbers (%). To evaluate differences between the two examinations at the start and end of fattening, calves (*n* = 600) were divided into two groups: TUS1 ^3^, TUS2 ^4^. Differences between healthy and ill calves were marked at a significance of *p* < 0.05.

		TUS1 ^3^ (*n* = 600)		TUS1 ^4^ (*n* = 600)	
Variable	Category	ILL ^1^ *n* = 336 (56.0) ^A,b^	HEALTHY ^2^ *n* = 264 (44.0) ^B,b^	ILL ^1^ *n* = 290 (48.3) ^a^	HEALTHY ^2^ *n* = 310 (51.7) ^a^
Frequency of treatments	0	256 (76.2) ^A,b^	259 (98.1) ^B,b^	227 (78.3) ^A,a^	288 (92.9) ^B,a^
	1	60 (17.9) ^A^	5 (1.9) ^B,b^	43 (14.8) ^A^	22 (7.1) ^B,a^
	2	14 (4.2) ^A^	0 (0.0) ^B^	14 (4.8) ^A^	0 (0.0) ^B^
	3	1 (0.3)	0 (0.0)	1 (0.3)	0 (0.0)
	4	5 (1.5) ^A^	0 (0.0) ^B^	5 (1.7)	0 (0.0)
	total	80 (23.9) ^A,b^	5 (1.9) ^B,b^	63 (21.6) ^A^	22 (7.1) ^B,a^
sex	male	123 (36.6) ^A,b^	68 (25.8) ^B,b^	101 (34.8) ^a^	90 (29.0) ^a^
	female	213 (63.4) ^A,b^	196 (74.2) ^B,b^	189 (65.2) ^a^	220 (71.0) ^a^
breed	purebred	143 (42.6)	101 (38.3)	132 (45.5) ^A^	112 (36.1) ^B^
	crossbred	193 (57.4)	163 (61.7)	158 (54.5) ^A^	198 (63.9) ^B^
ADG	Median (1st and 3rd Quartile)	0.96 (0.8; 1.09) ^A^	1.01 (0.88; 1.13) ^B^	0.94 (0.77–1.10)	1.01 (0.88–1.12)

^1^ Ultrasonographically ILL calves with a total score ≥ 4 points. ^2^ Ultrasonographically HEALTHY calves with a total score < 4 points. ^3^ Ultrasonographically ILL/HEALTHY calves of the first examination at the start of fattening (TUS1). ^4^ Ultrasonographically ILL/HEALTHY calves of the second examination at the end of fattening (TUS2). ^A,B^ Alphabetical upper case indices indicating significant differences between ILL (A) and HEALTHY (B) calves at *p* < 0.05. ^a,b^ alphabetical lower case indices indicating significant differences between ILL calves of TUS1 (a) and TUS2 (b) and significant differences between HEALTHY calves of TUS1 (a) and TUS2 (b).

**Table 3 animals-14-02311-t003:** Distribution of various parameters in *post mortem* inspection of the lungs. Results of ILL ^1^ and HEALTY ^2^ calves are presented in numbers (%). Differences between healthy and ill calves were marked at a significance of *p* < 0.05.

		PM ^3^ (*n* = 600)	
Variable	Category	ILL ^1^ *n* = 210 (35.0) ^A^	HEALTHY ^2^ *n* = 390 (65.0) ^B^
Frequency of disease-causing treatments	0	167 (79.5) ^A^	348 (89.2) ^B^
	1	28 (13.3)	37 (9.5)
	2	11 (5.2) ^A^	3 (0.8) ^B^
	3	0 (0.0)	1 (0.3)
	4	4 (2.0) ^A^	1 (0.3) ^B^
	total	43 (20.5)	42 (10.9)
sex	male	77 (36.7) ^A^	114 (29.2) ^B^
	female	133 (63.3) ^A^	276 (70.8) ^B^
breed	purebred	87 (41.4) ^A^	157 (40.3) ^B^
	cross-breed	123 (58.6) ^A^	233 (59.7) ^B^
ADG	Median (1st and 3rd Quartile)	0.91 (0.76–1.07)	1.01 (0.87–1.13)

^1^ Calves with lung lesions detected in *post mortem* inspection of the lungs (ILL). ^2^ Calves without lung lesions detected in *post mortem* inspection of the lungs (HEALTHY). ^3^ *Post mortem* inspection of the lungs (PM). ^A,B^ Alphabetical upper case indices indicating significant differences between ILL (A) and HEALTHY (B) calves at *p* < 0.05.

**Table 4 animals-14-02311-t004:** Distribution of calf characteristics and health status in 300 VAC ^1^ and 300 NVAC ^2^ calves on arrival at the fattening farm.

Variable	Category	VAC ^1^ (*n* = 300)	NVAC ^2^ (*n* = 300)
sex	male	93 (31.0)	98 (32.7)
	female	207 (69.0)	202 (32.7)
breed	purebred	116 (38.7) ^A^	172 (57.3) ^B^
	crossbred	184 (61.3) ^A^	128 (42.7) ^B^
Arrival weight (kg)	Median (1st and 3rd Quartile)	93 (83; 101)	88 (79; 96)
RS 1 ^4^	ILL	137 (45.7)	130 (43.3)
	HEALTHY	163 (54.3)	170 (56.7)
TUS 1 ^3^	ILL	160 (53.3)	176 (58.7)
	HEALTHY	140 (46.7)	124 (41.3)

^1^ Vaccination (VAC): intranasal injection of 2 mL (1 mL per nostril) of a commercially available multivalent vaccine against bovine respiratory syncytial virus (BRSV) and parainfluenza 3 (PI3) two days after arrival on the farm. ^2^ Negative Placebo-Group (NVAC): intranasal injection of 2 mL (1 mL per nostril) of a sterile saline solution two days after arrival at the farm. ^3^ Ultrasonographically (TUS) examined calves with a total score ≥ 4 points (ILL) or <4 points (HEALTHY). ^4^ Clinically examined (RS) calves with a total score ≥ 4 points (ILL) or <4 points (HEALTHY). ^A,B^ Alphabetical upper case indices indicating significant differences between VAC (A) and NVAC (B) calves at *p* < 0.05.

**Table 5 animals-14-02311-t005:** Difference of variables (ADG, disease frequency, fattening duration) in 600 VAC ^1^ and NVAC ^2^.

Variable	Category	VAC ^1^ (*n* = 300)	NVAC ^2^ (*n* = 300)
Frequency of treatments	0	266 (88.7) ^A^	249 (83.0) ^B^
	1	28 (9.3)	37 (12.3)
	2	6 (2.0)	8 (2.7)
	3	0 (0.0)	1 (0.3)
	4	0 (0.0) ^A^	5 (1.7) ^B^
	total	34 (11.3) ^A^	51 (17.0) ^B^
RS 2 ^4^	ILL	108 (36.0)	116 (38.7)
	HEALTHY	192 (64.0)	184 (61.3)
TUS 2 ^3^	ILL	142 (47.3)	148 (49.3)
	HEALTHY	158 (52.7)	152 (50.7)
Fattening duration (d)	Median (1st and 3rd Quartile)	76 (65; 86)	77 (69; 85)
ADG (kg/d)	Median (1st and 3rd Quartile)	0.98 (0.85–1.11)	0.97 (0.85–1.11)

^1^ Vaccination (VAC): intranasal injection of 2 mL (1 mL per nostril) of a commercially available multivalent vaccine against bovine respiratory syncytial virus (BRSV), infectious rhinotracheitis (IBR), parainfluenza 3 (PI3) two days after arrival at the farm. ^2^ Negative control (NVAC): intranasal injection of 2 mL (1 mL per nostril) of a sterile saline solution two days after arrival at the farm. ^3^ Ultrasonographically (TUS) examined calves with a total score ≥ 4 points (ILL) or < 4 points (HEALTHY). ^4^ Clinically examined (RS) calves with a total score ≥ 7 points (ILL) or < 7 points (HEALTHY). ^A,B^ Alphabetical upper case indices indicating significant differences between VAC (A) and NVAC (B) calves at *p* < 0.05.

## Data Availability

The data that support the findings of this study are available from the corresponding author upon reasonable request.

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
