# Peer review of "Associations between Ultrasonographically Diagnosed Lung Lesions, Clinical Parameters and Treatment Frequency in Veal Calves in an Austrian Fattening Farm"

_animals, 2024, doi:10.3390/ani14162311_

Round 1

Reviewer 1 Report

Comments and Suggestions for Authors

Dear Authors,

The manuscript titled "Associations between ultrasonographically diagnosed lung lesions, clinical parameters and treatment frequency in veal calves in an Austrian fattening farm" is well written with scientific soundness. But need to revise the keywords, with appropriate single words rather than long sentence.  

Manuscript is well organized according to the stated objective. Findings are consistent with the hypothesis. That is,  Study evaluated ultrasound and clinical health in veal calves, vaccination effects, growth, and production parameters.  Findings show host components contribute to BRD in veal calf production. Monitoring calf health and diagnostics can reduce consequences.  Discussion part is critically analyzed by previous references, and References are cited according to the subject discussed. Overall, I am satisfied with the findings observed with the stated hypothesis.

Author Response

Thank you very much for taking the time to review this manuscript. Please fint the detailed response to the re- submitted file below. 

Sincerely,

Julia Hoffelner

Reviewer 2 Report

Comments and Suggestions for Authors

The article titled "Associations between ultrasonographically diagnosed lung lesions, clinical parameters, and treatment frequency in veal calves in an Austrian fattening farm" addresses a very interesting and current topic. The authors conducted a solid study with a considerable amount of data, but the presentation of these data needs improvement and clarification. Currently, the article is confusing and unclear, making it difficult to understand the study's results, which are indeed very interesting. Below are my suggestions divided by section:

Introduction:

Expand the introductory section related to the problems associated with diagnosing BRD, noting that animals are often subclinical or have upper respiratory tract infections that do not require antibiotic treatment. Emphasize that there is no gold standard for in vivo diagnosis and expand the introduction on diagnostic methods with accuracy indices.

Other studies in various bovine production settings have used ultrasound for monitoring treatment and should be included in the bibliography.

Pay attention to abbreviations: they sometimes appear without being previously written out in full.

Materials and Methods:

The ethical committee's authorization is missing.

Animals and Housing: The paragraph provides enough information but is poorly organized. I recommend the authors first describe the normal routine and then detail the animals included in the study with inclusion and exclusion criteria, followed by the division into VAC and NVAC groups.

The section on scores in materials and methods is unclear. Did you use two scores? How was the auscultation score categorized? What do the various scores correspond to in auscultation?

Details on how the ultrasound was performed, including the ultrasound machine and probe used, are missing. Additionally, abbreviations should be spelled out when first used.

Was the score used at the slaughterhouse ever described and validated?

Was the statistical analysis performed using Excel?

Was the chi-square test used to examine the distribution of the variables sex, breed, and frequency of treatments among the groups or between the VAC and NVAC groups?

Line 187: Change "metric" to "quantitative variable."

Lines 195-197: Move the sample size information to the beginning of the materials and methods section and elaborate on it. Which test was used to determine the sample size? What was the chosen response variable? Since you have two treatment groups, how many calves needed to be included in each group at minimum? What power level did you choose?

The introduction presents a combined classification method using ultrasound and clinical score, but this is not used later. Why?

The materials and methods section is unclear, making the results difficult to understand.

Results:

How many calves were ultimately included? How many were vaccinated and how many were not? A description of the total sample is missing.

It is unclear how healthy and sick animals are defined. Why did the animals in the healthy group have ultrasound lesions? Clarify the classifications in materials and methods.

When discussing treatment, what do you mean? I believe you mean antibiotic treatments, but it is unclear whether you refer to vaccination and whether the treatments included are only for BRD or other diseases as well.

In paragraph 3.4, you state that there was no difference in ADG between ILL and HEALTHY calves, but in the next paragraph (3.5), you say a difference was demonstrated. Clarify this.

The entire results section is confusing; the term "treatment" is unclear regarding the type of treatment and the disease treated. Furthermore, it is unclear what is meant by ILL and HEALTHY and why animals were divided by clinical score and not by ultrasound score.

Line 394: Remove the period.

Discussion:

In the discussion, expand on the use of ultrasound and its limitations. The discussion should be adjusted according to the corrections made in the previous sections.

Author Response

Thank you very much for taking the time to review this manuscript. Please find the detailed chanches below in the re- submitted file. 

Sincerely,

Julia Hoffelner
